# Laser Surface Hardening of Carburized Steels: A Review of Process Parameters and Application in Gear Manufacturing

**DOI:** 10.3390/ma18153623

**Published:** 2025-08-01

**Authors:** Janusz Kluczyński, Katarzyna Jasik, Jakub Łuszczek, Jakub Pokropek

**Affiliations:** Institute of Robots and Machine Design, Faculty of Mechanical Engineering, Military University of Technology, 00-908 Warsaw, Poland; janusz.kluczynski@wat.edu.pl (J.K.); katarzyna.jasik@wat.edu.pl (K.J.); jakub.pokropek@student.wat.edu.pl (J.P.)

**Keywords:** laser heat treatment, microhardness, surface hardening, 21NiCrMo2 steel

## Abstract

This article provides a comprehensive overview of recent studies concerning laser heat treatment (LHT) of structural and tool steels, with particular attention to the 21NiCrMo2 steel used for carburized gear wheels. Analysis includes the influence of critical laser processing conditions—including power output, motion speed, spot size, and focusing distance—on surface microhardness, hardening depth, and microstructure development. The findings indicate that the energy density is the dominant factor that affects the outcomes of LHT. Optimal results, in the form of a high surface microhardness and a sufficient depth of hardening, were achieved within the energy density range of 80–130 J/mm^2^, allowing for martensitic transformation while avoiding defects such as melting or cracking. At densities below 50 J/mm^2^, incomplete hardening occurred with minimal microhardness improvement. On the contrary, densities exceeding 150–180 J/mm^2^ caused surface overheating and degradation. For carburized 21NiCrMo2 steel, the most effective parameters included 450–1050 W laser power, 1.7–2.5 mm/s scanning speed, and 2.0–2.3 mm beam diameter. The review confirms that process control through energy-based parameters allows for reliable prediction and optimization of LHT for industrial applications, particularly in components exposed to cyclic loads.

## 1. Introduction

The heat treatment of steel is a process that enables the attainment of desired mechanical and microstructural properties of the material. In recent years, steel heat treatment technology has advanced due to laser use, which allows for a more precise and controlled heating of selected areas of processed materials [1,2,3,4,5]. Heat treatment using a laser beam is a process in which a concentrated beam of laser light is used to deliberately heat the material, leading to the transformation of its structure in a controlled manner [6]. The laser beam is focused on the target point, allowing for precise control of the process and local heating of the material [7,8]. This type of treatment has found wide application in the metal industry as well as in other fields such as medicine and electronics. It is used to improve mechanical properties, including the strength, hardness, and ductility of materials, which is particularly important in the production of tools, machine components, and aircraft parts [9]. The use of a laser as a heat source makes it possible to achieve high accuracy, repeatability, and shorter processing times, which significantly affects the efficiency of production processes [10].

The LHT process allows for precise control over the phase transformations that occur in the material. In the case of steel, laser surface hardening involves rapid and localized heating of the surface layer above the martensitic transformation temperature, that is, above the austenitizing temperature, which typically ranges from 800 to 950 °C depending on the alloy composition. At this temperature range, the initial microstructure (e.g., ferritic or pearlitic) transforms into austenite. Then, due to the steep thermal gradient and rapid self-quenching after the laser beam passes, the austenite undergoes a martensitic transformation, resulting in a hard and brittle phase. Crucially, the laser heats only the surface layer, while the underlying core remains relatively cool. This enables the formation of a hardened surface with significantly increased wear resistance and surface hardness, while preserving the ductility and toughness of the core. This mechanism, based on controlled austenitization followed by rapid martensitic transformation, is the fundamental principle of laser hardening and finds particular application in the precision treatment of components such as gear wheels [11,12]

Apart from the aforementioned applications related to LHT, the laser beam is also used for the precise cutting and joining of materials, which is essential in the production of detailed components and structures. The process itself involves focusing the laser beam on the area of the material to be treated, providing energy that heats the material, monitoring and controlling the process and its parameters such as power, scanning speed, and energy density, as well as cooling and hardening the material after the process is complete [13,14]. Regardless of the application of the laser beam, the main parameters of the process that uses it are laser power, which defines the energy input transmitted to the workpiece; scanning speed, which influences the process rate and uniformity of heating; and energy density, which controls penetration depth and thermal effects [15]. A schematic drawing of the LHT process is presented in Figure 1.

The aim of this literature review is to identify and analyze key parameters of the laser heat treatment process for steel, with particular emphasis on the influence of laser power, scanning speed, spot diameter, and energy density on hardness, microstructure, and hardened layer depth. The review focuses on the possibilities of using laser treatment in the context of precision hardening of 21NiCrMo2 steel, previously subjected to carburizing. The analysis of available experimental data aims to determine optimal technological conditions that allow for the production of a high-quality hardened layer without the risk of melting or microstructural degradation. This review lays the groundwork for future investigations and the broader adoption of LHT in processing components subjected to heavy mechanical loading.

## 2. Methodology of the Review of the Literature

The literature review was conducted to identify and analyze the key parameters of the LHT process applied to structural and tool steels, with a particular emphasis on applications involving gear wheels made of 21NiCrMo2 steel previously subjected to carburizing.

In the first stage, a selection of scientific publications was carried out, including articles from peer-reviewed journals, conference proceedings, and selected review articles published between 2008 and 2024. Using a tool provided by the Web of Science database, we analyzed the number of articles available under the search term “Laser heat treatment-steel” during the period under consideration. The graph below (Figure 2) visualizes how this technique has gained popularity.

Over several years, between 2008 and 2020, the number of publications related to LHT increased almost 20 times. By 2024, the number of papers remained at more than 200. This demonstrates the significant interest in the LHT technique in various centers around the world and supports the claim that it offers significant application potential with measurable economic benefits. In this review, priority was given to publications that contain detailed numerical and descriptive data related to the laser treatment process, such as laser power, scanning speed, spot diameter, type of laser source, final hardness, and chemical composition of the steel. Initially, it was a database of around 100 of the most relevant studies.

In the next step, works were identified that met specific inclusion criteria, which included known chemical composition of the material, with particular emphasis on carbon content; detailed description of laser hardening parameters; measurement of final hardness and hardened layer depth; and a description of the equipment used. In total, I managed to obtain about 34 papers. On the other hand, studies on nonferrous materials and publications lacking data that enabled energy-related calculations were excluded. For each publication, data on laser power, scanning speed, and spot diameter were extracted.

These parameters were compiled into summary tables along with the carbon content of the steel analyzed. This approach enabled a direct comparison of the effects of laser treatment across different studies, regardless of the type of light source used and equipment configuration. The analysis also considered the relationships between microhardness and energy density values, as well as the influence of parameters on microstructure, the presence of phase transformations (e.g., ferrite-martensite), decarburization phenomena, and the formation of residual stresses. Particular attention was paid to studies on carburized steels used in machine elements such as gear transmissions, for which laser surface hardening may serve as an alternative to conventional through-hardening methods. The collected data allowed for the determination of laser processing parameter ranges that enable the formation of a hardened layer with properties exceeding those of conventional hardening while simultaneously limiting the thermal impact on the core of given steel.

## 3. Results

A wide range of research has addressed both theoretical modeling and experimental investigations related to laser-based steel processing. For example, Dobrzeniecki et al. [16] employed a high-power diode laser to improve wear resistance and mechanical performance in two variants of steel. Wang et al. [17] developed numerical simulations capable of optimizing laser hardening parameters by predicting the depth of the treated layer. Similarly, Kusuhara et al. [18] applied a finite difference approach in combination with a three-dimensional finite element model to examine temporal variations in heat flow during laser treatment. In another comparative study, the authors of [19] evaluated laser and induction hardening techniques on cylindrical 25CrNiMo steel samples, emphasizing the superior energy efficiency and adaptability of laser-based methods.

To improve the tribological and fatigue characteristics of bearing raceways, He et al. [5] conducted transformation laser hardening experiments on 42CrMo steel. Their work examined how laser power and scanning speed affect phase transformation depth. The authors constructed and validated a numerical model by integrating experimental findings, thereby enabling a predictive framework for understanding the interaction between laser input and material response. Based on these insights, orthogonal arrays were formulated to determine optimal processing conditions.

The practical benefits of such surface modifications were further explored in [20], where the authors evaluated the mechanical performance of conical press fit joints with and without laser-strengthened contact paths. Building on the concept of enhancing fatigue and load-bearing properties through laser treatment, as demonstrated by He et al. [5], the study analyzed the strain and stress distribution in the shaft journal and bushing during three loading stages: the joining process, post-pressing, and under applied torsional moment. The results confirmed that localized laser strengthening of contact surfaces significantly increases the load capacity and structural integrity of the joints, highlighting the broader applicability of laser hardening techniques in mechanical assemblies.

In [21], the authors examined whether decarburization occurs during laser surface hardening of AISI 420 martensitic stainless steel. For comparison, the results of surface hardening and decarburization were also investigated in a conventional furnace heating process (water quenching and air cooling) for the same steel material. The ThermoCalc software (https://thermocalc.com/) was employed to simulate the depth-dependent distribution of carbon in samples subjected to both laser and conventional furnace hardening. Heat treatment provides many advantages, including the ability to precisely control the location and depth of heating. In addition, the process can be very fast, which increases production efficiency, and is cost-effective, as the concentrated laser beam minimizes energy losses [22].

Gelaw et al. [23], in their research, conducted an analysis of the laser hardening process of Unimax tool stainless steel, which was developed for the production of injection mold components. The composition of the steel used is presented in Table 1.

Despite its rather typical composition—lacking nickel and containing only 5% chromium—the steel is still considered stainless. The subject of the study was a non-hardened sample of this steel, which exhibits a hardness of approximately 200 HV. Through conventional hardening methods, this steel can be hardened to 600 HV. Therefore, the objective of the study was to achieve at least the same hardness level as in the case of conventional hardening techniques, with the added advantages of minimizing the processing time, self-quenching properties, and localized heat input.

For the experiments, 500 W Nd: A YAG Lumonics laser (Lumonics, Ottawa, Canada) was used, implemented in an integrated 5-axis CNC (Computer Numerical Control) machining center Sauer-70/5 (DMG Mori Co., Ltd., Tokyo, Japan). The laser can be operated either manually or by means of an NC code. Due to internal losses, the actual laser power of 500 W was reduced to 450 W [5]. Although the laser can theoretically generate variable output power through computer control, it does not operate very stably at varying machine power levels. Therefore, the laser functions best at a maximum effective power, that of 450 W.

To optimize laser hardening parameters, a total of 48 trials were conducted using three scanning speeds (1.7, 2.5, and 6.7 m/min) and four laser spot diameters (2.164, 2.169, 2.288, and 2.412 mm). Each combination was repeated three times to minimize random errors. The precise parameters of the laser hardening process are shown in Table 2.

The low scanning speed and small laser spot size resulted in a correspondingly higher hardness (780 HV) and higher hardening depth. This is due to the fact that a smaller spot size yields a higher heat density and a lower scanning speed allows for sufficient time for interaction with the workpiece to raise the sample temperature. However, these parameters cannot be considered optimal as they consistently lead to melting of the sample, which causes undesirable surface deformation of the material.

Ultimately, the optimal solution for a sample that did not experience deformation produced a hardness of 650 HV, which is slightly higher than the value achieved through conventional hardening, but with a significant advantage in processing time when using laser hardening. The optimal processing parameters are presented in Table 3.

The study [24] focused on examining how laser heat treatment influences the properties of a modified AISI P20 steel, an alloy frequently used in manufacturing injection molding tools. The detailed elemental composition of the material is summarized in Table 4.

Laser heat treatment was carried out using a diode laser LDF 4500-60 (Laserline GmbH, Mülheim-Kärlich, Germany). The laser head was mounted on a six-axis robot and equipped with a pyrometer and a camera. The diameters of the laser beam were 15.2 mm and 7.2 mm, respectively, and the maximum laser beam power was 450 W. The experimental work was divided into four series in which different input parameters were adjusted and then regulated. The only constant parameter was the power of the laser beam. Process parameters are presented in Table 5.

The baseline microhardness of the modified AISI P20 steel was approximately 300 HV. As anticipated, laser heat treatment led to a significant increase in surface hardness across the heat-affected zone. Specifically, in the upper region of the samples, the values rose from around 300 to approximately 625 HV. At a depth of 1.0 mm and with a scanning speed of 10 mm/s, the hardness readings ranged from 576 to 588 HV, gradually returning to the base material levels (~300 HV) at a depth of 1.2 mm. When the scanning speed was increased to 15 mm/s, the hardened layer extended only to 0.8 mm, exhibiting hardness values between 588 and 608 HV, followed by a decrease in the initial value.

These findings confirm that the laser process improved hardness by approximately 108% on the surface relative to untreated steel. This improvement is attributed to the phase transformation from ferrite to martensite induced by rapid heating and cooling. Although the degree of hardening was similar in different tests, a scanning speed of 10 mm/s yielded a deeper modified zone compared to the higher speed of 15 mm/s.

The effect of laser treatment on the roughness of the AISI P20 steel surface was also observed. An increase in roughness occurred along the longitudinal direction of the laser scanning after the treatment, accompanied by a reduction in the ratio between longitudinal and transverse roughness.

The purpose of the study in [8] was to optimize crash resistance and impact energy absorption in steel sheets that underwent laser hardening. The authors focused on various microstructural design strategies to tailor the mechanical properties of the material to the requirements that truck components must meet. To this end, commercial 22MnB5 steel sheets with a thickness of 6.2 mm were used. The sheets were hot-rolled and initially exhibited a ferrite–pearlite microstructure. The chemical composition of steel is presented in Table 6.

Laser treatment was carried out using a TRUMPF VCSEL laser module (TRUMPF Photonic Components GmbH, Ulm, Germany) with a power of 2400 W. Process parameters are presented in Table 7.

The study demonstrated different microstructures in 22MnB5 thick steel sheets by applying various irradiation strategies. Laser treatment proved to be an effective method for achieving a microstructure with properties similar to bainite, which is particularly advantageous for components where precise control over the course of thermomechanical transformations within the material structure is difficult.

In [15], the authors investigated the laser hardening process of 14Cr17Ni2 steel (AISI 431) using a fiber laser beam. The chemical composition of the steel is presented in Table 8.

The laser processing used continuous-wave (CW) emission using a defocused beam characterized by a Gaussian intensity profile and a central wavelength of 1.07 μm. The beam produced a circular irradiation spot on the surface of the material. Notably, no absorptive coatings were applied to the sample surface prior to treatment. The sample surface was not oxidized and did not contain deep indentations that could affect the laser radiation absorption coefficient. Experimental hardening conditions were chosen to ensure that the sample surface temperature remained below 1600 K, a threshold set by the operational constraints of the available measurement equipment.

The calculation results were experimentally validated. For this purpose, straight-line scanning paths were applied to samples with a thickness of 20 mm. Prior to experiments, the samples were degreased and their surfaces were ground to remove oxides, indentations, or scratches to stabilize the radiation absorption conditions.

The laser surface modification was performed using a processing unit equipped with a continuous-wave ytterbium fiber laser (LS-16, 16 kW, IPG Photonics, Oxford, MA, USA), which emits at a wavelength of 1.07 μm. During the treatment, the surface temperature within the laser-affected zone was monitored using an optical pyrometer (CT Laser 3MH2-SF, Optris, Berlin, Germany), with the sensing point placed 1 mm from the beam center. This configuration allowed for an accurate measurement of hardening temperatures while avoiding surface melting, as appropriate for the tested steel grade. A full summary of processing parameters is provided in Table 9.

Four-pass laser hardening had a positive effect on the depth of the hardened layer by increasing it. Microhardness also increased. The average microhardness in the hardened layer was 322 HV2 at Qn = 0.28 kW·s/mm and 319 HV2 at Qn = 0.42 kW·s/mm. Figure 3 presents the hardness profile of the hardened samples.

The authors of [25] used laser treatment to improve the fatigue life of steel gears that had been subjected to carburizing processes. The samples used in this study were made of 4118 steel, the chemical composition is presented in Table 10.

The specimens underwent normalization and carburizing treatments, followed by machining into cylindrical shapes with dimensions of 40 mm in diameter and 15 mm in thickness, in accordance with ISO 4885 [26] and JIS B6914 [27] standards. A semiconductor laser was used in the study. The laser processing parameters applied during the experiments are presented in Table 11.

The most significant result of the study was the substantial increase in fatigue life after laser treatment. The number of cycles to failure increased by a factor of 3.8 compared to the untreated material, suggesting that laser treatment could be an effective method for repairing fatigue damage in mechanical components, especially those subjected to carburizing. This study is highly relevant from the perspective of mechanical parts regeneration and life extension. Moreover, the use of this technology can contribute to reducing environmental impact, as it enables the reuse of worn components and reduces the need for new parts to be manufactured.

The study also demonstrated that laser treatment had a significant effect on the microstructure of the material. It caused the grain refinement of austenite and restored the retained austenite phase in the hardened layer. These microstructural changes contributed to the improvement of the material’s mechanical properties.

Before laser treatment, the surface hardness of the carburized steel was approximately HV700 (up to a depth of 200 µm). However, the hardness gradually decreased, reaching HV550 at a depth of approximately 1.0 mm, indicating the presence of a thin carburized layer on the surface. After laser processing, a significant increase in surface hardness was observed. This increase reached approximately HV800 and was visible up to a depth of 180 µm below the surface. In the deeper layers of the material, that is, at depths of 250 µm and 500–600 µm—the hardness was found to be HV500 and HV600, respectively. At a depth of 900 µm, the hardness was close to that of the untreated carburized material.

Laser hardening significantly increased the surface hardness of the material, which contributed to improved mechanical properties and wear resistance. This is a crucial aspect in the context of the regeneration and extension of the useful life of mechanical components.

Wagh et al. [28] investigated the effect of the parameters of the laser hardening process on properties such as hardness and microstructure in Ck45 steel. The detailed chemical composition of the investigated steel is presented in Table 12.

In the study, a fiber laser with a power of 400 W and a wavelength of 1070 nm was used. Research focused on varying three main process parameters: laser beam power, laser scanning speed, and the distance between the laser head and the workpiece. Nine experimental sets with different combinations of these parameters were carried out, as presented in Table 13.

This study focused on three key parameters: laser beam power, which was examined at three levels: 210 W, 270 W, and 330 W. The results showed that higher laser power led to a deeper hardness layer on the surface of the Ck45 steel. The laser scanning speed, which varied at three levels: 1.0 mm/s, 8.5 mm/s, and 16.0 mm/s, had a significant effect on surface hardness, with lower scanning speeds resulting in deeper hardening. Three different distances between the laser head and the workpiece, 200 mm, 250 mm, and 300 mm, were also investigated. Although this distance had an effect on hardness, it was less pronounced compared to the other parameters.

The study demonstrated that the optimal laser processing parameters were a laser power of 330 W, a scanning speed of 1.0 mm/s, and a laser head-to-workpiece distance of 300 mm, resulting in a maximum surface hardness depth of 340.25 μm in Ck45 steel. The microstructure of the Ck45 steel samples revealed two distinct zones: the heat-affected zone and the base metal zone. A non-linear regression analysis allowed for the development of a predictive equation for the hardness depth as a function of the process parameters.

In [29], the authors reported on a study of laser treatment of EN24 steel and its effects on microhardness and other surface properties of the material. The chemical composition of the steel is presented in Table 14.

A 400 W continuous-wave fiber laser with a wavelength of 1070 nm was used to perform the laser treatment. Various process parameters were tested, including laser beam power (210, 225, and 300 W), laser scanning speed (1.0, 5.5, and 10.0 mm/s), and the stand-off distance between the laser head and the workpiece (175, 200, and 225 mm). The detailed process configuration is presented in Table 15.

The study demonstrated that the optimal laser processing parameters for EN24 steel were a laser beam power of 300 W, a scanning speed of 1.0 mm/s, and a laser head-to-workpiece distance of 175 mm. Research focused on the microhardness of samples, which quantifies the extent to which a material can oppose permanent surface damage under contact stress. Before laser treatment, the microhardness of the base metal was 208.2 HV0.3. After laser treatment, the microhardness of the samples ranged from 516.3 to 728.1 HV0.3. As the laser beam power increased from 210 to 300 W, the microhardness also increased. This effect resulted from the maximum surface heating, which led to the formation of a martensitic structure that significantly increased the hardness of the material.

The surface microhardness was shown to decrease with increasing laser scanning speed. A higher scanning speed reduced the concentration of heat energy per unit of time, translating into lower microhardness values. Microstructural analysis revealed that the laser-hardened layer consisted of uniformly distributed martensite plates with several undissolved carbides and retained austenite. This microstructure was a key factor influencing the microhardness of the material.

In summary, the experimental results demonstrated that laser treatment significantly improved the hardness of EN24 steel, which could be highly beneficial in applications where material strength is critical. Optimization of process parameters, such as laser power and scanning speed, allowed for the best microhardness results to be achieved, which is relevant for industry and applications requiring high-strength materials.

Frerichs et al. [30] investigated the effect of the laser hardening process on residual stresses, microstructure, and hardness in AISI 4140 steel (42CrMo4). The composition of the steel is presented in Table 16.

In the study, a Yb:YAG laser with an operating wavelength of 1030 nm was used. Various process parameters, such as laser power, scanning speed, and heat flux, were applied. The scanning speeds and heat fluxes ranged from 0.5 to 20 mm/s and from 10 to approximately 46 W/mm^2^, respectively. A specific optical setup with lenses was employed to generate a rectangular laser beam with a constant width. The laser hardening process involved direct direction of the laser beam onto the surface of the steel component. The use of shielding gas (argon) was necessary to minimize oxidation of the treated surface. Temperature measurements were taken during the process, both on the surface and inside the material, using a pyrometer and thermocouples. The applied processing parameters are presented in Table 17.

As a result of laser hardening, a significant increase in residual stresses was observed on the surface of the tested steel material. The residual stress values were below 40 MPa on the surface, suggesting the presence of concentrated stresses within the top layer. The laser hardening process affected the microstructure of the material, as evidenced by phase transformations. The hardness was one of the key parameters investigated during the laser hardening process. After treatment, the material achieved a hardness level ranging from 30 HRC to 47 HRC. This increase in hardness suggests that the laser hardening process significantly influenced the mechanical properties of the material, making it considerably harder than in its initial state.

In their study, the authors of [31] investigated the feasibility of the application of fiber lasers in heat treatment and evaluated their performance limits and process quality in the context of gear surface hardening. An LS-20 laser was used, with a 200 μm transport fiber diameter, a laser power output of 20 kW, an IPGFLWD50500/160 focusing head, and a KUKA kr60 robot. The primary purpose of this equipment was laser welding; however, in this study, it was applied for laser treatment. The material investigated was 5Kh2MNF steel and its chemical composition is presented in Table 18.

The process parameters were then selected and are presented in Table 19.

The penetration depth of the heat treated layer was strongly influenced by the processing speed. Lower scanning velocities resulted in deeper hardening zones, up to approximately 2 mm, while higher speeds reduced the effective depth to less than 1 mm. An overall increase in surface hardness was observed with increasing laser power and scanning speed. For instance, a combination of 70 mm/s speed and 20 kW power produced a maximum hardness of 775 HV0.1. At 6 mm/s, the maximum hardness value recorded was 700 HV0.1, while treatment at 3 mm/s and 5 kW yielded an average hardness of around 650 HV0.1.

In recent years, finite element modeling (FEM) has become an essential tool for predicting thermal fields, phase transformations, and hardness profiles in laser surface hardening. Its growing application significantly reduces experimental workload and enables the selection of optimal processing parameters through simulation rather than trial-and-error.

Recent numerical studies reinforce the value of FEM-based approaches in optimizing laser hardening of carburized steels. Wyględacz et al. [32] used Sysweld to model thermal cycles in laser hardening of WCL tool steel; simulations matched experimental temperature measurements at depths up to 2.0 mm, with peak surface temperatures exceeding 1100 °C. Zhang et al. [33] applied a hybrid FEM–Kriging model to optimize processing parameters for bearing steel, achieving surface hardness up to 730 HV and reducing the number of physical trials by over 60%. Łach [34] reviewed several FEM studies that predicted hardness depths ranging from 0.5 to 1.6 mm, with phase transformation zones modeled accurately under laser powers between 400 and 1500 W. Sun et al. [35] used FEM to simulate laser shock processing of 20CrMnTi steel, resulting in surface hardness increases from ~340 HV to over 520 HV and a refined compressed layer depth of approximately 200 µm.

The reviewed literature and numerical modeling results confirm that both careful selection of processing parameters and the use of modern simulation tools, such as FEM, are essential for designing an effective laser hardening process for carburized steels—ensuring high surface hardness, controlled case depth, and minimal thermal side effects.

## 4. Discussion

A literature review was conducted to determine the optimal parameters for the laser hardening process of gears made of 21NiCrMo2 steel that had undergone carburizing. Due to the use of different types of laser in the reviewed studies—resulting in variations in laser power and beam diameter—the energy density was calculated based on the parameters provided by the authors using the following Formula (1). This allowed for the selection of process parameters such as laser power, laser scanning speed, and stand-off distance between the laser head and the workpiece.(1)$$Laser\;energy\;density=\frac{laser\;power}{diameter\;of\;laser\;spot\cdot scanning\;speed}\left[\frac{J}{{mm}^{2}}\right]$$

The process parameters, including the percentage of carbon content in the investigated material and the calculated energy density, are presented in the tables included in Appendix A. Analysis of the compiled data and the hardness values reported in the respective studies indicates that the key factor determining the effectiveness of laser hardening of steel is the energy density delivered to the material. Optimal hardening results, in the form of high surface layer microhardness and appropriate hardening depth, were achieved at energy density values ranging approximately from 80 to 130 J/mm^2^. Within this range, stable martensitic structure formation was observed, along with a reduced risk of undesirable phenomena such as melting, decarburization, or microcracking.

At lower energy densities (<50 J/mm^2^), the hardening effect was incomplete: a full phase transformation did not occur and the resulting microhardness remained at the base material level, thereby showing only a slight increase. On the contrary, values exceeding 150–180 J/mm^2^, although locally increasing hardness, often led to overheating and surface degradation, manifested by material melting or geometry distortion.

Based on data from the literature for steels with carbon content similar (before carburizing) to that of 21NiCrMo2, it can be concluded that the most favorable results were obtained with laser powers ranging from 450 to 1050 W, scanning speeds between 1.7 and 2.5 mm/s, and a laser spot diameter from approximately 2.0 to 2.3 mm. This configuration ensured a balance between the intensity of the heating and the control over the hardening process.

The results of this analysis are graphically presented in Figure 4, which illustrates the relationship between energy density and the resulting microhardness. The plot includes data points collected from the literature [8,15,22,24,25,28,29,30,31], covering a wide spectrum of experimental conditions and materials. A third-degree polynomial regression was fitted to the data to highlight the trend of increasing hardness with energy density, up to a critical threshold beyond which material degradation begins. Three distinct process zones are indicated on the graph:The incomplete hardening zone (<50 J/mm^2^), characterized by low or insufficient microhardness;The optimal hardening zone (80–130 J/mm^2^), where stable martensitic transformation and high surface hardness were observed;The overheating zone (>150 J/mm^2^), in which excessive energy input resulted in partial melting or surface damage.

This graphical representation reinforces the conclusion that the effectiveness of the process in laser hardening can be accurately predicted and optimized through control of the energy density.

Although the review includes 30 carefully selected studies, this number may appear limited considering the broad scope of laser heat treatment applications. This relatively small sample size is primarily due to the stringent inclusion criteria used in this work, which required well-defined processing parameters, known chemical compositions, and quantifiable hardness results. Moreover, the body of literature specifically focused on laser surface hardening of carburized steels is still emerging.

As a result, the findings and recommendations presented here should be interpreted with an understanding of this limitation. While the analysis identifies general trends and optimal parameter ranges, the relatively modest data set may constrain the generalizability of the conclusions across all steel grades or laser system configurations. Future research expanding the dataset, particularly through standardized experimental protocols, will be crucial to validate and refine these findings.

To support the findings of the literature review, an experimental laser surface hardening was performed on samples made of 21NiCrMo2 steel (after carburizing). The samples were irradiated using a fiber laser, with processing parameters selected based on optimal energy densities in the range from 80 to 130 J/mm^2^. The resulting microstructural transformations within the laser-affected zone were examined using scanning electron microscopy (SEM). Representative results are shown in Figure 5A–C.

Figure 5A–C present a sequence of SEM images that illustrate the transformation of the microstructure from the laser-hardened path to the base material. Figure 5A shows a cross-sectional view of the hardened zone with a characteristic semi-elliptical profile. The surface exhibits a visibly refined microstructure, indicating a martensitic transformation as a result of rapid self-quenching during laser treatment. Figure 5B shows the microstructure of the base material, unaffected by the laser beam, with a typical ferritic–pearlitic morphology. Figure 5C, at higher magnification, reveals pearlite colonies, lacrima cementite plates embedded within a ferritic matrix, commonly observed in carburized steels. These observations confirm the formation of a hardened surface layer with a clear structural contrast relative to the untreated core. The visual evidence aligns with the literature findings describing microstructural refinement and phase transformation under sufficient energy input during laser hardening.

## 5. Conclusions

This article presents a review of the literature on laser heat treatment (LHT) of structural and tool steels, with a particular focus on the 21NiCrMo2 steel used in gear production. The analysis includes experimental research results on the influence of process parameters, such as laser power, scanning speed, beam diameter, and stand-off distance, on surface hardness, hardening depth, and the resulting microstructure of treated materials.

Based on empirical data collected from several dozen sources, summary tables of technological parameters and their corresponding energy values were compiled. Special attention was paid to the selection of parameters for 21NiCrMo2 steel that had been subjected to prior carburizing, with the aim of applying LHT technology for the hardening of gears exposed to cyclic fatigue loading.

The following conclusions were formulated on the analyzed data:Laser heat treatment enables effective and precise hardening of the surface layers of low-carbon steels (after prior carburizing) and alloy steels while maintaining the integrity of the core microstructure.The most favorable hardening results were achieved at energy density levels ranging from 80 to 130 J/mm^2^, which allowed the surface microhardness to exceed 600 HV without the risk of melting.An increase in laser power without an appropriate adjustment of scanning speed can lead to material degradation due to melting or decarburization.For 21NiCrMo2 steel, laser heat treatment can serve as an effective alternative to conventional post-carburizing hardening, offering better control of the heat-affected zone and reduced residual stresses.

Additionally, the following technological recommendations are proposed to guide experimental research on laser hardening of 21NiCrMo2 steel using model samples:The steel should be subjected to a carburizing process to obtain a high surface hardness.Recommended laser power: 450–1050 W;Recommended wavelength of 1070 nm;Scanning speed: 1.7 to 2.5 mm/s;Laser spot diameter: 2.1–2.3 mm;Energy density: 80–130 J/mm^2^;Standoff distance: 110 mm (with a standard collimator configuration).Excessively low scanning speeds and excessively high power values should be avoided, as they may lead to surface melting and distortion of part geometry.Preliminary hardness testing and microstructural analysis of the samples are recommended prior to implementing these parameters in production lines.To further enhance fatigue life, it is recommended to consider the use of multi-track hardening strategies or beam shaping techniques (e.g., ring-shaped spots, beam modulation), which enable more uniform temperature distributions.

## Figures and Tables

**Figure 1 materials-18-03623-f001:**
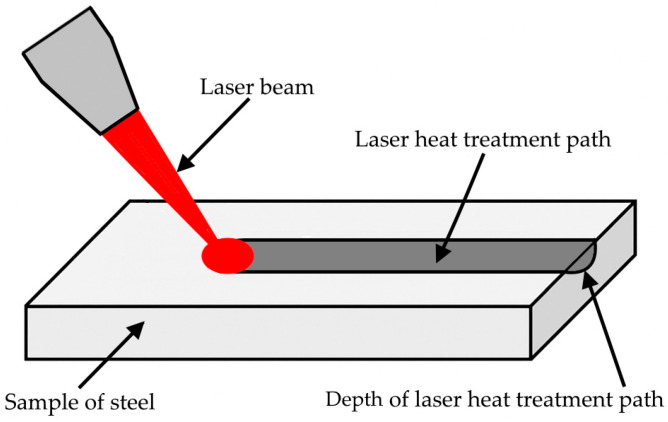
Graphical diagram of the LHT process.

**Figure 2 materials-18-03623-f002:**
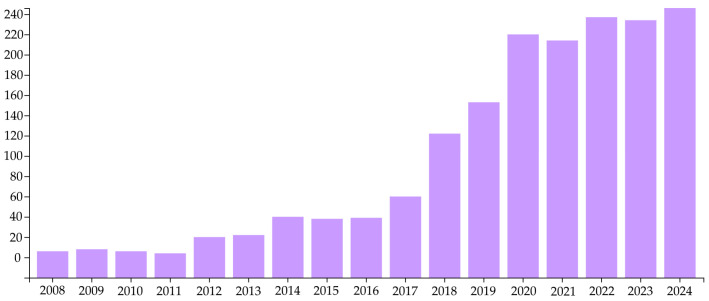
Number of publications on LHT from 2008 to 2024. Data from the Web of Science database.

**Figure 3 materials-18-03623-f003:**
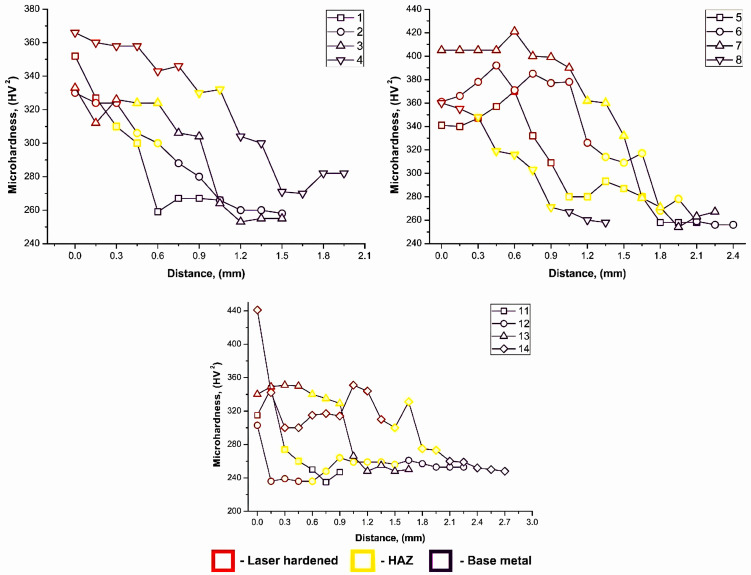
Hardness profiles of the laser-hardened AISI 431 samples [15].

**Figure 4 materials-18-03623-f004:**
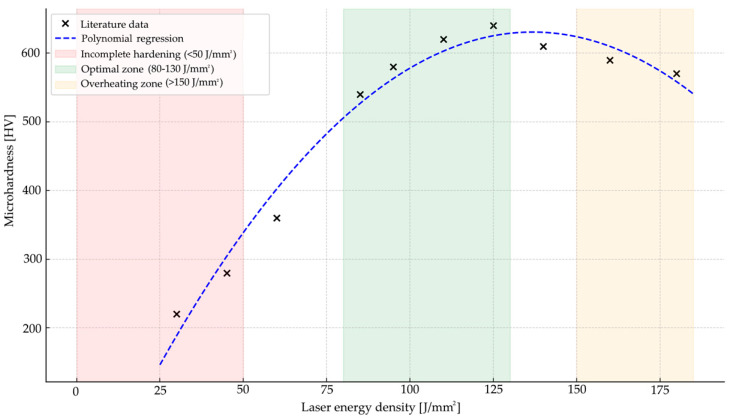
Effect of energy density on surface microhardness on literature data. The chart includes a polynomial regression curve and highlights three characteristic process zones: incomplete hardening (<50 J/mm^2^), optimal hardening (80–130 J/mm^2^), and overheating (>150 J/mm^2^). Data compiled from studies [8,15,22,24,25,28,29,30,31].

**Figure 5 materials-18-03623-f005:**
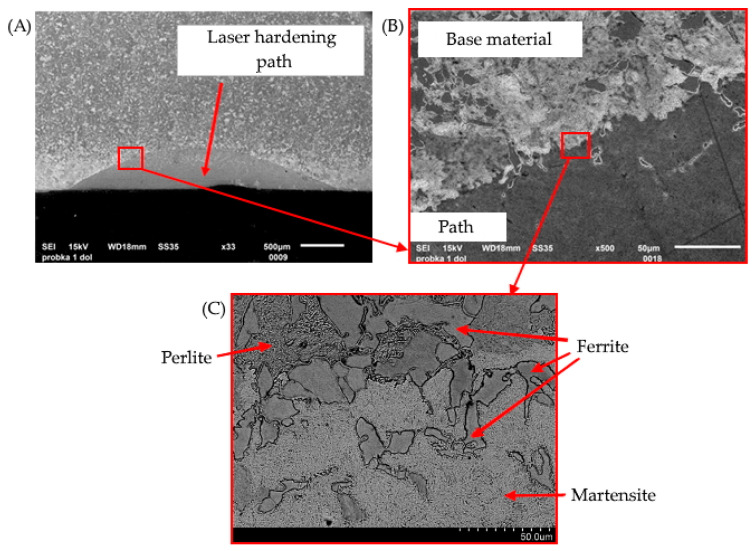
Microstructural evolution in 21NiCrMo2 steel after laser surface hardening: (**A**) cross-section of the laser-hardened path, showing surface refinement, (**B**) base material with ferritic–pearlitic microstructure, unaffected by the laser, (**C**) high-magnification SEM image of pearlite colonies (lamellar cementite in ferrite matrix)—own research.

**Table 1 materials-18-03623-t001:** The chemical composition of Unimax steel [23].

Chemical Element	C	Si	Mn	Cr	Mo	V
Elemental Content [%]	0.5	0.2	0.5	5.0	2.3	0.5

**Table 2 materials-18-03623-t002:** Parameters of laser treatment [5].

Number of Group	Laser Power [W]	Scanning Speed [m/min]	Diameter of Laser Spot [mm]	Distance Between Laser Head and Treatment Element [mm]
1	450	1.7	2.164	105
2	450	1.7	2.169	110
3	450	1.7	2.288	115
4	450	1.7	2.412	120
5	450	2.5	2.164	105
6	450	2.5	2.169	110
7	450	2.5	2.288	115
8	450	2.5	2.412	120
9	450	6.7	2.164	105
10	450	6.7	2.169	110
11	450	6.7	2.288	115
12	450	6.7	2.412	120

**Table 3 materials-18-03623-t003:** Optimal processing parameters [5].

Laser Power [W]	Scanning Speed [m/min]	Diameter of Laser Spot [mm]	Distance Between Laser Head and Treatment Element [mm]
450	2.5	2.169	110 mm

**Table 4 materials-18-03623-t004:** The chemical composition of the investigated AISI P20 steel [24].

Chemical Element	C	Si	Mn	Cr	Mo	Ni
Elemental Content [%]	0.40	0.30	1.45	1.95	0.2	1.05

**Table 5 materials-18-03623-t005:** Parameters of laser treatment of AISI P20 steel [24].

Number of Group	Laser Power [W]	Scanning Speed [m/min]	Diameter of Laser Spot [mm]
1	450	10	15.2
2		15	7.2
3		10	7.2
4		15	15.2

**Table 6 materials-18-03623-t006:** The chemical composition of the investigated of 22MnB5 steel [8].

Chemical Element	C	Si	Mn	Cr	Al	B
Elemental Content [%]	0.215	0.27	1.19	0.119	0.03	0.02

**Table 7 materials-18-03623-t007:** Parameters of laser treatment of 22MnB5 steel [8].

Laser Power [W]	Scanning Speed [m/min]	Diameter of Laser Spot [mm]	Surface Temperature [°C]
2400	0.6	15.2	930

**Table 8 materials-18-03623-t008:** The chemical composition of the investigated AISI 431 steel [15].

Chemical Element	C	Si	Mn	Cr	Al	Ni	S	P	Cu
Elemental Content [%]	0.14	≤0.8	≤0.8	17	0.03	2	≤0.025	≤0.03	≤0.3

**Table 9 materials-18-03623-t009:** Parameters of laser treatment of AISI 431 steel [15].

Number of Group	Laser Power [W]	Heat Input [kW∙s/mm]	Diameter of Laser Spot [mm]	Surface Temperature [K]
1	16,000	0.22	8	1473
2	16,000	0.4	8	1373
3	16,000	0.43	8	1523
4	16,000	0.46	8	1522

**Table 10 materials-18-03623-t010:** Chemical composition of 4118 steel [25].

Chemical Element	C	Si	Mn	Mo	Cr
Elemental Content [%]	0.2	0.22	0.8	0.15	0.5

**Table 11 materials-18-03623-t011:** Parameters of laser treatment of 4118 steel [25].

Laser Power [W]	Scanning Speed [m/min]	Diameter of Laser Spot [mm]	Surface Temperature [°C]
1050	1.525	1	930

**Table 12 materials-18-03623-t012:** The chemical composition of the investigated Ck45 steel [28].

Chemical Element	C	Si	S	P	Mn
Elemental Content [%]	0.54	0.32	0.039	0.035	0.76

**Table 13 materials-18-03623-t013:** Parameters of laser treatment of Ck45 steel [28].

Number of Group	Laser Power [W]	Scanning Speed [m/min]	Distance Between Laser Head and Treatment Element [mm]
1	210	1	200
2	210	8.5	250
3	210	16	300
4	270	1	250
5	270	8.5	300
6	270	16	200
7	330	1	300
8	330	8.5	200
9	330	16	250

**Table 14 materials-18-03623-t014:** The chemical composition of EN24 steel [29].

Chemical Element	C	Si	S	P	Mn	Cr	Ni	Mo
Elemental Content [%]	0.386	0.228	0.028	0.014	0.542	1.364	1.392	0.24

**Table 15 materials-18-03623-t015:** Parameters of laser treatment of EN24 steel [29].

Number of Group	Laser Power [W]	Scanning Speed [m/min]	Distance Between Laser Head and Treatment Element [mm]
1	210	1	175
2	210	5.5	200
3	210	10	225
4	255	1	200
5	255	5.5	225
6	255	10	175
7	300	1	225
8	300	5.5	175
9	300	10	200

**Table 16 materials-18-03623-t016:** The chemical composition of the investigated AISI 4140 steel [29].

Chemical Element	C	Cr	Mo	Si	Mn
Elemental Content [%]	0.43	1.09	0.25	0.26	0.74

**Table 17 materials-18-03623-t017:** Parameters of laser treatment of AISI 4140 steel [29].

Number of Group	Laser Power [W]	Scanning Speed [m/min]	Diameter of Laser Spot [mm]	Heating Time [s]	Surface Temperature [°C]
1	165	0.03	1.48	2.960	1100
2	264	0.06	1.48	1.480	1100
3	323	0.12	1.48	0.749	1100
4	385	0.24	1.48	0.370	1100
5	490	0.48	1.48	0.185	1100
6	555	0.72	1.48	0.123	1100
7	625	0.96	1.48	0.925	1100
8	680	1.2	1.48	0.074	1100

**Table 18 materials-18-03623-t018:** The chemical composition of the investigated 5Kh2MNF steel [31].

Chemical Element	C	Si	Mn	Ni	S	P	Cr	Mo	V	Cu
Elemental Content [%]	0.46–0.53	0.1–0.4	0.4–0.7	1.2–1.6	≤0.03	≤0.03	1.5–2.0	0.8–1.1	0.3–0.5	≤0.3

**Table 19 materials-18-03623-t019:** Parameters of laser treatment of 5Kh2MNF steel [31].

Number of Group	Laser Power [W]	Scanning Speed [m/min]	Diameter of Laser Spot [mm]
1	20,000	70	34
2	5000	6	34
3	5000	5	43
4	5000	3	43
5	5000	5	34

## Data Availability

No new data were created or analyzed in this study. Data sharing is not applicable to this article.

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
