# Peer review of "Laser Surface Hardening of Carburized Steels: A Review of Process Parameters and Application in Gear Manufacturing"

_materials, 2025, doi:10.3390/ma18153623_

Round 1

Reviewer 1 Report

Comments and Suggestions for Authors

The paper lists some references for laser heat treatment of carburized steel. However, it does not give scientific information on the process, how this process works and the influence on the microstructure and property evolution. 

Author Response

Dear Reviewer,

In the beginning, we would like to thank you for your revision and comments, which were very helpful to improve our work. All corrections made in our manuscript, which were made based on your comments, have been purple-highlighted. Below you can find our answer to your comment.

"The paper lists some references for laser heat treatment of carburized steel. However, it does not give scientific information on the process, how this process works, and the influence on the microstructure and property evolution."

Response:

We thank the Reviewer for this valuable comment. We fully agree that the scientific foundation of the laser heat treatment (LHT) process, including the physical mechanisms and their effect on microstructure and mechanical properties, is essential to the completeness of this review.

In response, we have implemented the following revisions in key sections of the manuscript:

1. In the Introduction, we added a detailed explanation of the laser hardening mechanism. This includes the localized heating of the steel surface above the austenitizing temperature, rapid self-quenching, and subsequent martensitic transformation. We also described how the thermal gradient between surface and core enables surface hardening without affecting core ductility.

2. In the Discussion section, we expanded the analysis of the influence of laser parameters—especially energy density—on the transformation mechanisms and resulting properties. We distinguish three characteristic zones:

* Incomplete hardening (<50 J/mm²),

* Optimal hardening (80–130 J/mm²),

* Overheating (>150 J/mm²).

These zones are correlated with changes in surface hardness, microstructure, and potential defects such as melting or decarburization.

3. In the Discussion, we also included results from scanning electron microscopy (SEM) of 21NiCrMo2 steel samples (after carburizing and laser hardening). The micrographs clearly illustrate the transformation from the ferritic–pearlitic base microstructure to a martensitic surface layer and support the findings from the literature.

4. To visually support the process explanation, we added a new schematic diagram illustrating the LHT setup. The figure shows the laser head emitting a beam focused

on the steel surface, generating a hardened track. Key elements of the system are labeled for clarity, including the laser source, beam path, hardened zone, and movement direction. This figure is intended to assist readers less familiar with the LHT process.

5. In the Conclusions, we revised the summary to emphasize the predictive value of energy density in controlling the process and achieving consistent hardening effects in carburized steels such as 21NiCrMo2.

We believe that these comprehensive additions significantly strengthen the manuscript and directly address the Reviewer’s concerns.

Reviewer 2 Report

Comments and Suggestions for Authors

This review analyzes existing research on laser heat treatment (LHT) of structural and tool steels, with a focus on carburized 21NiCrMo2 steel used in gear wheels. The study evaluates the influence of critical process parameters (i.e. laser power, scanning speed, beam diameter, and focal distance) on surface microhardness, hardening depth, and microstructural changes. The literature confirms that energy density is the most significant factor influencing LHT outcomes. Optimal performance was observed within an energy density range of 80–130 J/mm².

The contribution is well-written and surely of interest for the readership of Materials. There are a couple of points that can be imporved, therefore my recommendation is publication after minor revision.

Comments:

  1. Avoid repeating the definition of the acronym on line 78, as it has already been introduced earlier in the text.
  2. Please specify in the text the number of scientific contributions and technical reports considered at each stage of the methodology (Section 2), including initial screening, eligibility, and final inclusion. This will improve the transparency and clarity of your review process.
  3. The review includes only 30 references, which is relatively limited for a literature review of this scope. While the scarcity of available data in the field is understandable, the manuscript should explicitly acknowledge this limitation and discuss how it impacts the reliability and generalizability of the findings.
  4. Given that only three combinations are presented for each parameter, is it necessary to include Table 2 in full? Consider whether a more concise description in the text may be sufficient, or if the table can be simplified.
  5. Consider adding micrographs to the manuscript to visually support your discussion of microstructural evolution and phase transformations resulting from laser heat treatment. These would greatly enhance the clarity and impact of your analysis.

Author Response

Dear Reviewer,

In the beginning, we would like to thank you for your revision and comments, which were very helpful to improve our work. All corrections made in our manuscript based on your comments have been yellow-highlighted. Below you can find our answer to your comments.

1. Avoid repeating the definition of the acronym on line 78, as it has already been introduced earlier in the text.

Response 1: The repetition has been removed; only the acronym is used.

2. Please specify in the text the number of scientific contributions and technical reports considered at each stage of the methodology (Section 2), including initial screening, eligibility, and final inclusion. This will improve the transparency and clarity of your review process.

Response 2: The number of studies considered at each stage has been added.

3. The review includes only 30 references, which is relatively limited for a literature review of this scope. While the scarcity of available data in the field is understandable, the manuscript should explicitly acknowledge this limitation and discuss how it impacts the reliability and generalizability of the findings.

Response 3: A remark regarding the impact of the number of included articles on the reliability of the findings and the generalizability of conclusions has been added to the “Summary” section.

„Although the review includes 30 carefully selected studies, this number may appear limited considering the broad scope of laser heat treatment applications. This relatively small sample size is primarily due to the stringent inclusion criteria used in this work, which required well-defined processing parameters, known chemical compositions, and quantifiable hardness results. Moreover, the body of literature specifically focused on laser surface hardening of carburized steels is still emerging. As a result, the findings and recommendations presented herein should be inter-preted with an understanding of this limitation. While the analysis identifies general trends and optimal parameter ranges, the relatively modest dataset

may constrain the generalizability of the conclusions across all steel grades or laser system configurations. Future research expanding the dataset, particularly through standardized experimental protocols, will be crucial to validate and refine these findings.”

4. Given that only three combinations are presented for each parameter, is it necessary to include Table 2 in full? Consider whether a more concise description in the text may be sufficient, or if the table can be simplified.

Response 4: This has been simplified into the following text: “To optimize the laser hardening parameters, a total of 48 trials were conducted using three scanning speeds (1.7, 2.5, and 6.7 m/min) and four laser spot diameters (2.164, 2.169, 2.288, and 2.412 mm). Each combination was repeated three times to minimize random error.”

5. Consider adding micrographs to the manuscript to visually support your discussion of microstructural evolution and phase transformations resulting from laser heat treatment. These would greatly enhance the clarity and impact of your analysis.

Response 5: Microstructural images of the steel, originating from our own research, have been added to the discussion. We believe this is a valuable addition to the “Discussion” section.

Reviewer 3 Report

Comments and Suggestions for Authors

The manuscript submitted for review deserves high praise, although the topic of the work is quite well-known and widespread, however, analysis is never superfluous.

There are several comments on the text:

1 The meaning of the sentence on line 57 – 58 is not clear.

2 It would be interesting to analyze the number of publications on this topic for the specified years 2008-2024, how much popularity and prevalence of this hardening method has grown, for which materials it is mainly used, etc., as it is done in the work Review: Laser welding of dissimilar materials (Al/Fe, Al/Ti, Al/Cu) – methods and techniques, microstructure and properties.

3 The quality of Figure 1 is unacceptable.

4 The recommended laser wavelength is not specified on lines 462 – 473.

Author Response

Dear Reviewer,

In the beginning, we would like to thank you for your revision and comments, which were very helpful to improve our work. All corrections made in our manuscript based on your comments have been green-highlighted. Below you can find our answer to your comments.

1. The meaning of the sentence on line 57 – 58 is not clear.

Response 1: The sentence has been corrected. The entire paragraph has been edited.

2. It would be interesting to analyze the number of publications on this topic for the specified years 2008-2024, how much popularity and prevalence of this hardening method has grown, for which materials it is mainly used, etc., as it is done in the work Review: Laser welding of dissimilar materials (Al/Fe, Al/Ti, Al/Cu) – methods and techniques, microstructure and properties.

Response 2: An analysis of the number of articles from 2008 to 2024 has been implemented. I have also added a chart presenting the data graphically.

3. The quality of Figure 1 is unacceptable.

Response 3: The image quality has been improved.

4. Given that only three combinations are presented for each parameter, is it necessary to include Table 2 in full? Consider whether a more concise description in the text may be sufficient, or if the table can be simplified.

Response 4: This has been simplified into the following text: “To optimize the laser hardening parameters, a total of 48 trials were conducted using three scanning speeds (1.7, 2.5, and 6.7 m/min) and four laser spot diameters (2.164, 2.169, 2.288, and 2.412 mm). Each combination was repeated three times to minimize random error.” - highlighted in yellow because the same comment was from another reviewer.

5. The recommended laser wavelength is not specified on lines 462 – 473.

Response 5: Recommended laser wavelength was added to technological recommendations.

Round 2

Reviewer 1 Report

Comments and Suggestions for Authors

The revised version can be accepted after correcting some grammatical mistakes.

Author Response

The whole manuscript was subjected to additional proofreading. Minor grammatical and stylistic corrections have been applied throughout the manuscript, including correction of typographical errors and improved phrasing in several sections. Everything is marked by the track changes tool.